# Urinary Dickkopf-3 (DKK3) Is Associated with Greater eGFR Loss in Patients with Resistant Hypertension

**DOI:** 10.3390/jcm12031034

**Published:** 2023-01-29

**Authors:** Ann-Kathrin C. Schäfer, Dennis Pieper, Hassan Dihazi, Gry H. Dihazi, Stephan Lüders, Michael J. Koziolek, Manuel Wallbach

**Affiliations:** 1Department of Nephrology and Rheumatology, University Medical Center Göttingen, 37075 Göttingen, Germany; 2Center for Biostructural Imaging of Neurodegeneration (BIN), University Medical Center Göttingen, 37075 Göttingen, Germany; 3Institute for Clinical Chemistry, University Medical Center Göttingen, 37075 Göttingen, Germany; 4Department of Nephrology, St. Josefs-Hospital, 49661 Cloppenburg, Germany; 5German Center for Cardiovascular Research (DZHK), Partner Site Göttingen, 37075 Göttingen, Germany

**Keywords:** Dickkopf-3, resistant hypertension, chronic kidney disease

## Abstract

Patients with resistant hypertension (HTN) demonstrate an increased risk of chronic kidney disease and progression to end-stage renal disease; however, the individual course of progression is hard to predict. Assessing the stress-induced, urinary glycoprotein Dickkopf-3 (uDKK3) may indicate ongoing renal damage and consecutive estimated glomerular filtration rate (eGFR) decline. The present study aimed to determine the association between uDKK3 levels and further eGFR changes in patients with resistant HTN. In total, 31 patients with resistant HTN were included. Blood pressure and renal function were measured at baseline and up to 24 months after (at months 12 and 24). uDKK3 levels were determined exclusively from the first available spot urine sample at baseline or up to a period of 6 months after, using a commercial ELISA kit. Distinctions between different patient groups were analyzed using the unpaired t-test or Mann–Whitney test. Correlation analysis was performed using Spearman’s correlation. The median uDKK3 level was 303 (interquartile range (IQR) 150–865) pg/mg creatinine. Patients were divided into those with high and low eGFR loss (≥3 vs. <3 mL/min/1.73 m²/year). Patients with high eGFR loss showed a significantly higher median baseline uDKK3 level (646 (IQR 249–2555) (n = 13) vs. 180 (IQR 123–365) pg/mg creatinine (n = 18), *p* = 0.0412 (Mann–Whitney U)). Alternatively, patients could be classified into those with high and low uDKK3 levels (≥400 vs. <400 pg/mg creatinine). Patients with high uDKK3 levels showed significantly higher eGFR loss (−6.4 ± 4.7 (n = 11) vs. 0.0 ± 7.6 mL/min/1.73 m^2^/year (n = 20), *p* = 0.0172 (2-sided, independent t-test)). Within the entire cohort, there was a significant correlation between the uDKK3 levels and change in eGFR at the latest follow-up (Spearman’s r = −0.3714, *p* = 0.0397). In patients with resistant HTN, high levels of uDKK3 are associated with higher eGFR loss up to 24 months later.

## 1. Introduction

Patients with resistant hypertension (HTN) demonstrate an increased risk of chronic kidney disease (CKD) and progression to end-stage renal disease (ESRD) [1,2,3]; however, the individual course of progression is difficult to predict using the established markers for CKD progression, such as estimated glomerular filtration rate (eGFR) and albuminuria. Patients within the same KDIGO risk category based on these parameters have a highly variable course in their kidney disease progression [4]. Even prediction tools with multiple parameters do not fit for all CKD patients [5]. It is also possible that renal function remains stable without evidence of CKD progression [4]. Therefore, it is of special interest to identify patients with ongoing CKD progression to optimize individual therapy.

Tubulointerstitial fibrosis and tubular atrophy are the histopathological characteristics of advanced CKD and decreasing kidney function [6,7]. In the development of CKD, tubular epithelial cells play a crucial role, driving inflammation and renal fibrosis [8]. Biomarkers detecting fibrotic processes and indicating even subclinical changes constitute a promising approach toward individualized assessment of CKD progression [9,10]. Thus, the urinary glycoprotein Dickkopf-3 (DKK3), whose secretion occurs stress-induced by tubular epithelial cells [11], might represent an interesting protein which may provide insights into ongoing tubulointerstitial fibrosis. DKK3 has profibrotic properties, e.g., modulating the Wnt/β-catenin signaling pathway, stimulating the expression of transforming growth factor-β (TGF-β), influencing local T-cell response, activating myofibroblasts and driving epithelial-to-mesenchymal transition [11,12,13,14,15]. Since DKK3 is expressed in tubular cells after injury, urinary DKK3 (uDKK3) might serve as a marker for ongoing tubular stress and CKD progression [11,16]. Accordingly, there is evidence that measurement of uDKK3 represents a novel tool for the identification of patients at high risk of eGFR decline with various subtypes of CKD [16]. 

In this context, assessment of the uDKK3 level may improve the management of patients with resistant HTN to halt CKD progression. However, whether the uDKK3 level might indicate further eGFR decline in patients with resistant HTN is unknown at present. Therefore, the present study aimed to determine the association between uDKK3 levels and further eGFR decline in patients with resistant HTN.

## 2. Materials and Methods

### 2.1. Study Design

Patients fulfilling the diagnostic criteria of resistant HTN with a blood pressure (BP) above national and international targets were evaluated and treated as described previously [17]. In particular, patients who had the combination of the following criteria were included: (1) office systolic BP (SBP) ≥ 140 mmHg, in general, or ≥130 mmHg for patients with CKD and proteinuria, as confirmed by duplicate measurements, despite maximal tolerated and optimized therapy, and (2) prescribed at least 3 antihypertensive medications including a diuretic. Patients were excluded if they suffered from CKD 5D. The present analysis included 31 patients treated from 06/2012 to 08/2016. Patients were part of a previous published study investigating the effect of baroreflex activation therapy (BAT) on 24 h BP [17]. BP and renal parameters were measured at baseline and up to 24 months after implantation of the BAT device. For analysis, the latest follow-up values for BP and eGFR were used (evaluation time: months 12 and 24). The median time of the latest follow up was 24 (IQR 24–24) months. Evaluated parameters were gender, age, body mass index (BMI), history of diabetes mellitus or smoking, number of prescribed antihypertensive drugs, office and ambulatory BP, renal function and uDKK3 concentration. All patients provided written informed consent before study initiation. The study complies with the principles of the Declaration of Helsinki and the local ethical committee approved the study protocol (# 19/9/11). 

### 2.2. Blood Pressure Measurement 

For office readings, BP measurement was performed on both upper arms. The arm with the higher value was used for all following measurements. Subsequently, BP was measured twice within a 3 min interval using a semiautomatic oscillometric device (Bosch und Sohn GmbH u. Co. KG, Jungingen, Germany) after 10 min of patient rest. The results of the two readings were averaged. The 24 h ambulatory BP (ABP) was investigated using an oscillometric Spacelabs Model 90207 Recorder (Spacelabs Healthcare GmbH, Nürnberg, Germany) with measurements every 15 min during the day and every 30 min at night. Readings were averaged after 24 h. Office BP as well as ABP were obtained from the last follow-up, conducted up to 24 months after baseline measurement. 

### 2.3. Assessment of uDKK3 and Renal Function 

Urine samples were prepared essentially as described previously [18]. Preanalytical urine samples were collected and centrifuged at 1000× *g* for 10 min at 4 °C to remove cell debris and casts. All urine samples were then immediately frozen at −80 °C. The uDKK3 levels were determined exclusively from the first available spot urine sample at baseline or up to a period of 6 months later (median 0 months (IQR 0–3)) in duplicate, using a commercial ELISA kit (ReFiNE Dkk3 ELISA, DiaRen UG, Homburg (Saar), Germany) according to the manufacturer’s instructions (https://www.diaren.de/fileadmin/user_upload/PDFs/Dkk3_ELISA_Dosier_ReFiNE_Testkit_Manual_DeV2017_06_25.pdf, accessed on 7 January 2023) and as described previously [16]. The uDKK3 concentrations were normalized to urinary creatinine concentrations to consider dilution of the urine. According to the manufacturer’s information, the coefficient of variation for intra-assay variability of repeated urine sample measurements is 3.1% in the lower detection range (at 488 pg DKK3/mL urine) and 3.5% in the upper detection range (at 1472 pg DKK3/mL urine). The values for inter-assay test variability are 4.7% in the lower detection range and 5.1% in the higher detection range. For the present analysis, all samples were measured on one ELISA plate.

The eGFR value was determined using the CKD EPI formula. Based on other studies [19] and Youden´s index, patients were divided into those with high and low eGFR loss (≥3 vs. <3 mL/min/1.73 m^2^/year) and into those with high and low uDKK3 levels (≥400 vs. <400 pg/mg creatinine). Correlation analyses of uDKK3 levels, albuminuria and baseline eGFR with changes in eGFR were performed.

### 2.4. Statistics

Data analysis was performed using the statistical software GraphPad Prism 9 and Microsoft Excel 2010. The D´Agostino and Pearson omnibus normality test was used to test data for a normal distribution. Differences in the investigated variables at different time points were investigated using the paired t-test or Wilcoxon signed-rank test. Categorical variables were compared using the chi-squared test. Distinctions between different patient groups were analyzed using the unpaired t-test or the Mann–Whitney test. For determination of the best discriminating cut-off uDKK3 concentration, Youden´s index was used. Correlation analysis was performed using Spearman’s correlation. Results are expressed as the mean value ± standard deviation (SD) or median and interquartile range (IQR). The threshold for statistical significance was chosen to be *p* < 0.05.

## 3. Results

### 3.1. Baseline Characteristics

Baseline characteristics and eGFR categories at baseline are presented in Table 1. The median uDKK3 level was 303 (IQR 150–865) pg/mg creatinine.

To achieve the best discrimination accuracy between patients with and without CKD progression, Youden´s index for the cut-off uDKK3 concentration was determined. A uDKK3 level of 398 pg/mg creatinine showed the highest discrimination accuracy between patients with and without eGFR decline at latest follow-up (Youden´s index 0.61, sensitivity 61.1%, specificity 100%). For better distinctiveness, this value was rounded up to 400 pg/mg creatinine.

In comparison to the baseline characteristics of patients with a uDKK3 level above (n = 11) or below (n = 20) 400 pg/mg creatinine, patients with higher uDKK3 levels showed a significantly reduced baseline eGFR (58 ± 35 vs. 80 ± 23 mL/min/1.73 m², *p* = 0.0429) and elevated albuminuria (1117 (38–2901) vs. 15 (10–39) mg/g creatinine, *p* = 0.0004). They also took significantly fewer antihypertensive drugs (5.9 ± 1.7 vs. 7.1 ± 1.3, *p* = 0.0464) without differences in the intake of nephroprotective ACE inhibitors or AT1 antagonists. Detailed information is expressed in Table 1.

The uDKK3 level correlated significantly with baseline eGFR (Spearman’s r = −0.4527, *p* = 0.0106) and the baseline urine albumin–creatinine ratio (UACR, Spearman’s r = 0.6415, *p* = 0.0001), whereas it did not correlate with baseline office or ambulatory BP.

### 3.2. Development of Blood Pressure and Renal Function

Among all patients, office systolic BP distinctly declined from 171 ± 23 mmHg to 148 ± 22 mmHg (*p* = 0.0001) by the time of latest follow-up. Of the 31 analyzed patients, 4 patients (13%) had their latest follow-up with measurement of eGFR and BP at month 12, and 27 patients (87%) had their latest follow-up at month 24. There were no significant differences in the office and ambulatory BP changes in patients with uDKK3 levels above or below 400 pg/mg creatinine (Table 2).

Patients with uDKK3 levels ≥400 pg/mg creatinine showed a significantly higher loss of eGFR from baseline to last follow-up compared to patients with lower uDKK3 levels (−6.4 ± 4.7 vs. 0.0 ± 7.6 mL/min/1.73 m^2^/year, *p* = 0.0172, see Figure 1). Differences in relative eGFR decline between these two groups were even higher (−13.4 ± 8.1% in patients with baseline uDKK3 level ≥400 pg/mg creatinine and −0.7 ± 12.2% in patients with baseline uDKK3 level <400 pg/mg creatinine, *p* = 0.0042).

Moreover, in patients with an average eGFR loss of ≥3 mL/min/1.73 m^2^/year (n = 13), significantly higher uDKK3 levels were measured in the first 6 months than in patients with less annual eGFR decline (<3 mL/min/1.73 m^2^/year) (646 (IQR 249–2555) vs. 180 (IQR 123–365) pg/mg creatinine, (n = 18) *p* = 0.0412, Figure 2). There was no significant correlation of baseline eGFR with the change in eGFR at month 12 (r = −0.041, *p* = 0.38) and month 24 (r = 0.183, *p* = 0.36).

Within the cohort, there was a significant correlation between the baseline uDKK3 level and the absolute change in eGFR (mL/min/1.73 m^2^) at the latest follow-up (Spearman’s r = −0.3714, *p* = 0.0397) as well as with the change in eGFR per year (Spearman’s r = −0.3750, *p* = 0.0376). The correlations between the baseline uDKK3 level and percentage change in eGFR at the latest follow-up and between the baseline uDKK3 level and relative change in eGFR per year were even stronger (Spearman’s r = −0.4791, *p* = 0.0064 and −0.4682, *p* = 0.0079).

In contrast, baseline eGFR and UACR only correlated significantly with the percentage decline in eGFR and not with the absolute change in eGFR (see Table 3).

In patients with normal eGFR and albuminuria less than 30 mg/g creatinine (n = 8), the median baseline DKK3 level was 143 (IQR 23–345) pg/mg creatinine. These patients showed a stable eGFR with a mean decline in eGFR of 0.88 ± 6.4 mL/min/1.73 m²/year and percentage decline of 0.83 ± 6.6% per year.

Table 4 shows eGFR categories at baseline and the latest follow-up in respect to baseline uDKK3. Deterioration of eGFR categories from baseline to the latest follow-up occurred in 6 patients (55%) with DKK3 ≥ 400 pg/mg creatinine and in 4 patients (20%) with DKK3 < 400 pg/mg creatinine (*p* = 0.049).

## 4. Discussion

In the present study, patients with resistant HTN with elevated uDKK3 levels ≥ 400 pg/mg creatinine showed significantly higher eGFR loss up to 24 months later compared to patients with lower uDKK3 levels. A significant correlation between uDKK3 levels at baseline and changes in eGFR at the latest follow-up at a median of 24 months later (IQR 24–24) was observed. These results are in line with previous reports showing that higher uDKK3 levels are associated with greater eGFR decline over the following 12 months in patients with CKD of different etiologies [16] or even in a non-CKD cohort [20]. However, the predictive value of the uDKK3 level for the long-term CKD prognosis, defined as the occurrence of ESRD or 40–50% eGFR decrease, is unknown at present.

As DKK3 is present in the developing kidney, then downregulated in adults and once again upregulated under pathological conditions within the kidney, elevated uDKK3 levels indicate ongoing tubular stress [11]. Furthermore, DKK3 promotes the development of and correlates with the extent of interstitial fibrosis and tubular atrophy, which are the hallmarks of progressive kidney disease [11]. In contrast, other promising renal biomarkers, such as NGAL, KIM-1, calprotectin and [TIMP2]*[IGFBP7], failed to show CKD progression in IgA nephropathy [21]. Additionally, higher albuminuria is rather uncommon in hypertensives, and proteinuria is often unimpressive or absent in HTN-related CKD [22]. However, a substantial eGFR decline may also be present in patients with non-proteinuric CKD, suggesting that measurement of albuminuria might also be less reliable for progression prediction. Accordingly, in the present study, only baseline uDKK3 correlated with absolute and percentage eGFR decline, whereas baseline eGFR and albuminuria were associated with only percentage eGFR decline. Accordingly, the correlation of uDKK3 level with the percentage change in eGFR was stronger than that of UACR. Therefore, to our current knowledge, uDKK3 may be most suitable to indicate and graduate the risk of CKD progression in patients with resistant HTN. This provides an opportunity to identify patients who may need and benefit from a stricter therapy setting.

As higher uDKK3 levels are associated with acute kidney injury (AKI) after potential nephro-aggressive interventions and also with AKI-to-CKD transition [23,24,25], the uDKK3 level may serve as a useful risk marker for identifying patients who would benefit from nephroprotective therapies, especially in vulnerable cohorts, as in patients with resistant HTN.

There was a numerically higher decrease in systolic office and ambulatory BP in patients with uDKK > 400 pg/mg creatinine compared to patients with uDKK3 level <400 pg/mg creatinine without reaching statistical significance. It is possible that this tendency to a greater BP reduction as a result of BAT in patients with higher uDKK3 levels was caused by the fewer prescribed baseline antihypertensive medications taken by these patients. Nevertheless, it would be of interest to evaluate whether BP-lowering therapies might influence uDKK3 levels according to their nephroprotective effects. The numerically greater BP reduction in patients with higher uDKK3 levels after a median follow-up of 24 months (IQR 24–24) might have contributed to a reduction in eGFR decline in this group. Considering the nephroprotective effects of this greater BP reduction, the predictive power of the uDKK3 level might actually even be “masked” by this imbalance in BP control between the two groups.

Due to the limited availability of paired urine samples, the effect of the BAT device on uDKK3 levels could not be determined in the present study. Additionally, a larger cohort and a longer observation period would be required to determine this. Other limitations of this study were its single-center, non-randomized design and the different time periods at which follow-up measurements were collected. Although of interest, long-term CKD progression until development of ESRD was not evaluated in this study. Furthermore, an intrinsic propensity for CKD progression, due to reduced baseline eGFR in patients with baseline uDKK3 ≥ 400 pg/mg creatinine, independent of uDKK3 in the present patients, cannot be fully excluded. However, the lack of correlation between baseline eGFR and change in eGFR makes this rather unlikely. Moreover, recently published results demonstrate that even after adjustment for baseline eGFR, uDKK3 remained an independent indicator of further eGFR decline [16]. Compared with eGFR or albuminuria alone, the assessment of urinary DKK3 significantly improved the prediction of further eGFR decline [16]. Through substantial glycosylation, DKK3 reaches a molecular weight of 60–70 kDa [26], which makes an extensive glomerular filtration of DKK3 within an intact glomerulum unlikely. Although of special interest, the question of whether high uDKK3 in patients with low eGFR may also be a result of accumulation of DKK3 in blood, raising the urinary “load”, cannot be definitively answered at the current time. Future studies are necessary to explore the detailed route of urinary and plasma DKK3 during kidney injury.

## 5. Conclusions

In patients with resistant HTN, high uDKK3 levels are associated with a more pronounced future eGFR decline. In particular, patients with uDKK3 levels ≥400 pg/mL, in comparison to patients with baseline uDKK3 levels <400 pg/mL, showed a statistically significant difference in eGFR decline. This might help to identify patients with higher risk of CKD progression who would benefit from stricter therapy settings and nephroprotective therapies. Larger studies are needed to confirm this result and to test whether BP-lowering and nephroprotective therapies might influence uDKK3 levels through their nephroprotective effects. As the uDKK3 level might identify patients at high risk of CKD progression, it is of interest whether intensification of nephroprotective therapy dependent on uDKK3 levels might reduce CKD progression. Further studies are needed to investigate whether uDKK3 may serve as a biomarker to improve the management of patients with various kidney diseases. Moreover, studies investigating the association between uDKK3 and the development of renal endpoints would be of special interest to determine whether the uDKK3 level can predict a relevant clinical course.

## Figures and Tables

**Figure 1 jcm-12-01034-f001:**
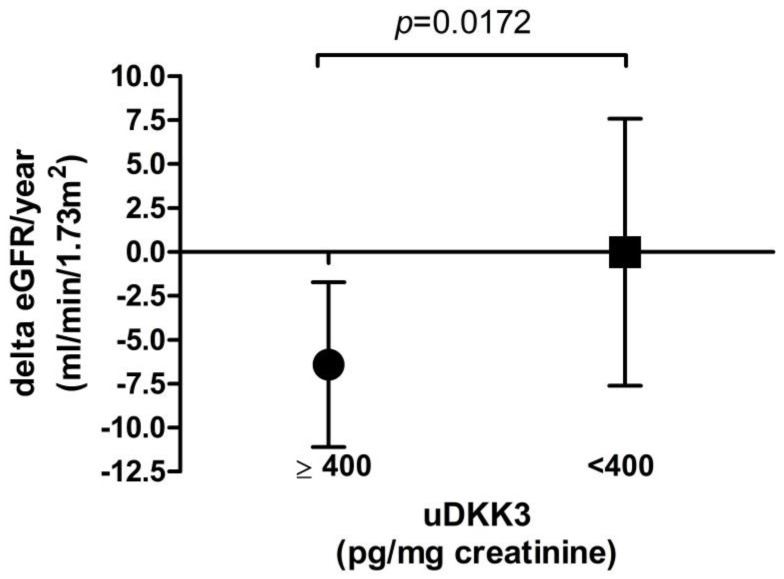
Absolute eGFR decline in patients with a baseline uDKK3 level above or below 400 pg/mg creatinine respective.

**Figure 2 jcm-12-01034-f002:**
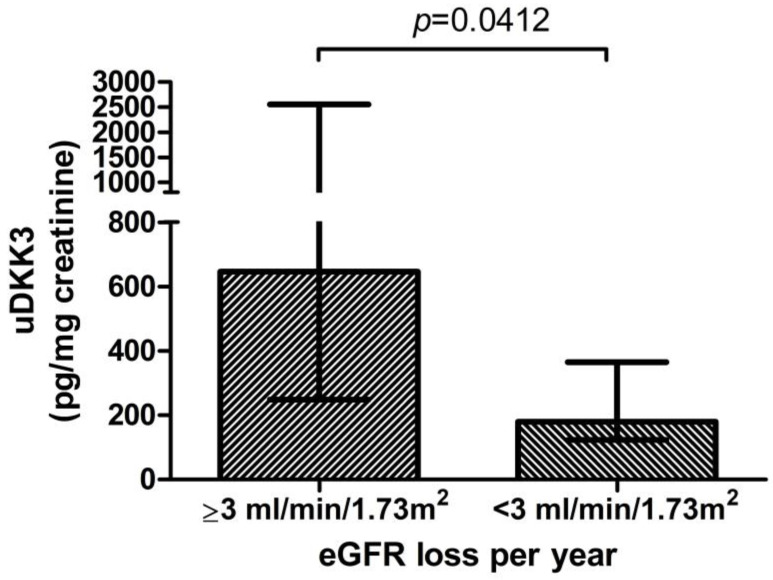
Baseline uDKK3 level in patients with or without 3 mL/min/1.73 m² annual loss of eGFR.

**Table 1 jcm-12-01034-t001:** Baseline characteristics ^#^.

Parameter	All Patients (n = 31)	Patients with Baseline uDKK ≥ 400 pg/mg Creatinine (n = 11)	Patients with Baseline uDKK < 400 pg/mg Creatinine (n = 20)	** *p* ** **-Value**
Age (years)	58 ± 13	62 ± 13	56 ± 13	0.12
Gender (female/male)	13/18	5/6	8/12	0.79
Mean BMI (kg/m^2^)	33 ± 6	34.9 ± 6.3	32.6 ± 6.0	0.47
Diabetes mellitus	16 (52%)	6 (55%)	10 (50%)	0.83
History of smoking	20 (65%)	8 (73%)	12 (60%)	0.56
Mean office BP (mmHg)	171 ± 23/90 ± 19	170 ± 29/92 ± 21	166 ± 18/89 ± 18	0.14/0.66
Mean ABP (mmHg)	148 ± 13/81 ± 11	151 ± 11/80 ± 7	147 ± 13/81 ± 12	0.43/0.92
Mean number of antihypertensive drugs	6.6 ± 1.5	5.9 ± 1.7	7.1 ± 1.3	0.0464 *
Median uDKK3 level (pg/mg creatinine)	303 (150–865)	1637 (745–3960)	162 (116–292)	<0.0001 *
Mean eGFR (mL/min/1.73 m^2^)	72 ± 29	58 ± 35	80 ± 23	0.0429 *
Median UACR (mg/g creatinine)	28 (14–643)	1117 (38–2901)	15 (10–39)	0.0004 *
CKD stage > 3	4 (13%)	4 (36%)	0 (0%)	0.0039 *

^#^ ABP—ambulatory blood pressure, BMI—body mass index, BP—blood pressure, CKD—chronic kidney disease, eGFR—estimated glomerular filtration rate, UACR—urine albumin–creatinine ratio, uDKK3—urinary Dickkopf-3. Values are expressed as mean ± standard deviation, number (%) or median and interquartile range. *p*-values are based on the comparison of patients with an uDKK3 level above or below 400 pg/mg creatinine. * marks significant values.

**Table 2 jcm-12-01034-t002:** BP decline in patients with baseline uDKK3 level above or below 400 pg/mg creatinine ^#^.

Parameter	Patients with Baseline uDKK ≥ 400 pg/mg Creatinine (n = 11)	Patients with Baseline uDKK < 400 pg/mg Creatinine (n = 20)	*p*-Value
Delta systolic office BP (mmHg)	−29.4 ± 24.0	−18.8 ± 25.6	0.28
Delta diastolic office BP (mmHg)	−6.3 ± 17.1	−5.2 ± 15.1	0.87
Delta systolic 24 h ABP (mmHg)	−11.0 ± 23.7	−3.7 ± 21.3	0.43
Delta diastolic 24 h ABP (mmHg)	−6.1 ± 11.2	−2.1 ± 14.0	0.36

^#^ ABP—ambulatory blood pressure, BP—blood pressure, uDKK3—urinary Dickkopf-3. Values are expressed as mean ± standard deviation. *p*-values are based on the comparison of patients with an uDKK3 level above or below 400 pg/mg creatinine.

**Table 3 jcm-12-01034-t003:** Correlation analysis of baseline uDKK3 level and change in eGFR ^#^.

Correlation	Baseline uDKK3 Level		Baseline eGFR		Baseline UACR	
	Spearman’s r	*p*-Value	Pearson’s r	*p*-Value	Spearman’s r	*p*-Value
Delta eGFR (mL/min/1.73 m^2^) at latest follow-up	−0.3714	0.0397 *	0.1524	0.4129	−0.2590	0.1669
Delta eGFR (mL/min/1.73 m^2^) per year	−0.3750	0.0376 *	0.1296	0.4872	−0.2659	0.1555
Percentage change in eGFR at latest follow-up	−0.4791	0.0064 *	0.4714	0.0074 *	−0.4014	0.0279 *
Percentage change in eGFR per year	−0.4682	0.0079 *	0.4110	0.0216 *	−0.3866	0.0348 *

^#^ eGFR—estimated glomerular filtration rate, UACR—urine albumin–creatinine ratio, uDKK3—urinary Dickkopf-3. Median time of latest follow-up was 24 (interquartile range 24-24) months. * marks significant values.

**Table 4 jcm-12-01034-t004:** Last follow-up changes of eGFR categories in respect to baseline uDKK3 ^#^.

	Baseline			Latest Follow-Up		
eGFR Categories (mL/min/1.73 m^2^)	Patients with Baseline uDKK ≥ 400 pg/mg Creatinine (n = 11)	Patients with Baseline uDKK < 400 pg/mg Creatinine (n = 20)	*p*-Value *^°^*	Patients with Baseline uDKK ≥ 400 pg/mg Creatinine (n = 11)	Patients with Baseline uDKK < 400 pg/mg Creatinine (n = 20)	*p*-Value *^+^*
>90	2 (18%)	7 (35%)	0.0304 *	1 (9%)	8 (40%)	0.0808
60–89	4 (36%)	8 (40%)		3 (27%)	6 (30%)	
30–59	1 (9%)	5 (25%)		2 (18%)	5 (25%)	
15–29	4 (36%)	0 (0%)		3 (27%)	1 (5%)	
<15	0 (0%)	0 (0%)		2 (18%)	0 (0%)	
Decrease in eGFR category	-	-		6 (55%)	4 (20%)	0.049 *

^#^ eGFR—estimated glomerular filtration rate, uDKK3—urinary Dickkopf-3. Values represent n (%). *p*-values are based on the comparison of patients with an uDKK3 level above or below 400 pg/mg creatinine (^°^ at baseline, ***^+^*** at latest follow-up). * marks significant values.

## Data Availability

The data presented in this study are available on request from the corresponding author. The data are not publicly available as patients did not agree for their data to be shared publicly.

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
