# Peer review of "Urinary Dickkopf-3 (DKK3) Is Associated with Greater eGFR Loss in Patients with Resistant Hypertension"

_jcm, 2023, doi:10.3390/jcm12031034_

Round 1

Reviewer 1 Report

Overall, this is a very interesting study and below please find some recommendations to improve it. 

Title: It should be more specific to the research question/aims

Abstract: The abstract needs major editions. 

-The authors should mention the study design and aim.  it should be clear the follow up period / data collection times that will lead to understand better the analysis 

-The statistical methods should be mentioned to clarify the results obtained within each category. The authors mentioned categories classification, eGFR loss and correlation in the entire cohort but none previous definitions/methods were provided, same for abbreviations 

- Line 23, the median values should appear earlier, before the classification categories

Introduction: 

Material and Methods: 

-The authors should explain briefly the definition of resistant HTN 

-Blood pressure measurement section should clarify the follow-up times for assessment of each one, BP and ABP

-Urine samples data collection times are not specified, only the window time of collection. The authors should mention if only one sample was taken or more than that, mentioning briefly the methods that were published previously, wherever relevant.

- The methods section does not provide information about the frequency of variables assessment, loss to follow up or other characteristics of this study design. 

Results:

- Line 119 suggests again a comparison against the baseline but none follow-up was mentioned. 

- The results are interesting and the editions in the methods section should allow the reader to follow them better 

Discussion:

- Line 174 mentions a "short-term" loss that should be defined in the introduction, considering that the follow-up median time is 24 months but none range was provided.  

-Line 176 should be edited to specify if the authors wanted to refer to uDKK3 baseline levels and eGFR of follow-ups. 

-Line 210 mentions important limitations that should be also mentioned in the study design, such as follow-up times range, frequency of assessment/samples 

Conclusions

-The authors mentioned a conclusion about association but, so far, they only mentioned a correlation analysis. Please clarify (Line 97-103 of methods) 

-Line 220, please mention other analysis that you would recommend to perform not only limited to sample size 

Author Response

Reviewer 1

Overall, this is a very interesting study and below please find some recommendations to improve it. 

  1. Title: It should be more specific to the research question/aims

Answer: We thank the reviewer for this suggestion. We specified the title within the revised version of the manuscript.

“Urinary Dickkopf-3 (DKK3) is associated with greater eGFR loss in patients with resistant hypertension”

Abstract: The abstract needs major editions. 

  1. The authors should mention the study design and aim. It should be clear the follow up period / data collection times that will lead to understand better the analysis 

Answer: We extended information for the study design and the aim of the study within the revised version of the abstract.

(…The present study aimed to determine the association between uDKK3 levels and further eGFR changes in patients with resistant HTN….)

 (…31 patients with resistant hypertension were included in this analysis. Blood pressure and renal function were measured at baseline and up to 24 months (at months 12 and 24). DKK3 levels were determined exclusively once within the first available spot urine at baseline or up to a period of 6 months using a commercial ELISA kit….)

  1. The statistical methods should be mentioned to clarify the results obtained within each category. The authors mentioned categories classification, eGFR loss and correlation in the entire cohort but none previous definitions/methods were provided, same for abbreviations 

Answer: We thank the reviewer for the suggestion of extending definition and methods. Within the revised abstract, we briefly extended relevant definitions/methods and abbreviations.

  1. Line 23, the median values should appear earlier, before the classification categories

Answer: According to the reviewer’s suggestion, we included median DKK3-level earlier in the abstract before introducing to the classification categories.

Introduction: 

Material and Methods: 

  1. The authors should explain briefly the definition of resistant HTN 

Answer: We agree with the reviewer’s suggestion, that definition of resistant HTN helps the reader to better understand the manuscript. We included a brief explanation within the revised method section.

(…In particular, patients who had the combination of the following criteria were included (1) office systolic BP (SBP) ≥140 mm Hg in general or ≥130 mm Hg for patients with CKD and proteinuria, confirmed by duplicate measurements, despite maximal tolerated and optimized therapy with at least 3 antihypertensive medications including a diuretic…)

  1. Blood pressure measurement section should clarify the follow-up times for assessment of each one, BP and ABP

Answer: We thank the reviewer for this suggestion. We added the requested information to the revised version of the manuscript.

Methods

(…Follow-up times were month 12 and 24. For the present analysis (office BP and ABP), the latest follow-up values for each patients up to 24 months were selected for analysis….)

Results

(…In particular, of the 31 analyzed patients, 4 patients (13%) had latest follow-up with de-termination of eGFR and BP at months 12 and 27 patients (87%) at months 24….)

  1. Urine samples data collection times are not specified, only the window time of collection. The authors should mention if only one sample was taken or more than that, mentioning briefly the methods that were published previously, wherever relevant.

Answer: We thank the reviewer for the suggestion of further clarification of the methods section. We agree that as the present patients were part of prior analyses, the window time of collection is misleading. We clarified this within the revised version of the manuscript. Moreover, according to the reviewer’s suggestion, we extended briefly the methods wherever relevant.

Methods

 (...Urine samples were prepared essentially as described previously [18]. Preanalytical, urine samples were collected, centrifuged at 1,000 g for 10 min at 4 °C to remove cell debris and casts. Consecutively, all urine samples were immediately frozen at -80°C. uDKK3 levels were determined exclusively once within the first available spot urine at baseline or up to a period of 6 months in duplicate, using a commercial ELISA kit (ReFiNE Dkk3 ELISA, DiaRen UG, Homburg (Saar), Germany), according to manufactures instructions (https://www.diaren.de/fileadmin/user_upload/PDFs/Dkk3_ELISA_Dosier_ReFiNE_Testkit_Manual_DeV2017_06_25.pdf, downloaded 7th January 2023) and described before [16]. uDKK3 concentrations were normalized to urinary creatinine concentrations to consider dilution of the urine….)

  1. The methods section does not provide information about the frequency of variables assessment, loss to follow up or other characteristics of this study design. 

Answer: We thank the reviewer for this deep critical analysis of the manuscript. According to this suggestion, we add the requested information to the revised method section. As indicated within the methods section, with a median of 24 months (IQR 24-24) the vast majority of patients had latest follow-up at months 24. For the present analysis, BP and eGFR were evaluated in follow-up at month 12 and month 24 and the latest available follow-up value was used for evaluation.

Methods

(… For analysis, the latest follow-up values for BP and eGFR were used (evaluation time month 12 and 24…)

(…uDKK3 levels were determined exclusively once within the first available spot urine at baseline or up to a period of 6 months (median 0 months (IQR 0-3)) in duplicate,…)

Results

(…In particular, of the 31 analyzed patients, 4 patients (13%) had latest follow-up with determination of eGFR and BP at months 12 and 27 patients (87%) at months 24….)

Results:

  1. Line 119 suggests again a comparison against the baseline but none follow-up was mentioned. 

Answer: As indicated under point 7 and 8 we included information on follow-up time within the revised version of the manuscript.

  1. The results are interesting and the editions in the methods section should allow the reader to follow them better 

Discussion:

  1. Line 174 mentions a "short-term" loss that should be defined in the introduction, considering that the follow-up median time is 24 months but none range was provided.  

Answer: We thank the reviewer for this suggestion. The term short term seems to be misleading in this context, as patients’ last follow-up was at a median of 24 months (IQR 24-24) and the vast majority of patients (87%) had latest follow-up at month 24. We therefore removed the term “short-term” and specified the follow-up period where it is indicated.

  1. Line 176 should be edited to specify if the authors wanted to refer to uDKK3 baseline levels and eGFR of follow-ups. 

Answer: According to reviewer’s suggestion, we specified the correspondent sentence within the revised version of the manuscript.

Discussion

(…A significant correlation between uDKK3 levels at baseline and changes of eGFR at the latest follow-up at a median of 24 months (IQR 24-24) was observed…)

  1. Line 210 mentions important limitations that should be also mentioned in the study design, such as follow-up times range, frequency of assessment/samples 

Answer: We thank the reviewer for this suggestion. Within reviewer’s point 8 we included follow-up time range, frequency of assessment within the study design (method section) in the revised version of the manuscript.

Conclusions

  1. The authors mentioned a conclusion about association but, so far, they only mentioned a correlation analysis. Please clarify (Line 97-103 of methods) 

Answer: We agree with the reviewer that only Spearmen correlation is a rather weak hint to conclude an association. However, further analysis were performed, justifying the term “association” between uDKK3 levels and change in eGFR at latest follow-up. In addition to the Spearmen correlation, analysis showing a correlation between uDKK3-levels and eGFR decline at latest follow-up, comparison of groups of patients with uDKK3 levels ≥400 pg/ml and <400pg/ml showed a statistically significant difference in eGFR drop (Figure 1). Vice versa patients with eGFR-loss per year ≥3 ml/min/1.73m2 and <3ml/min/1.73m2 showed statistically significant differences (Figure 2). As indicated by the reviewer we clarified this within the revised version of the manuscript.

Conclusion

(…In patients with resistant HTN, high uDKK3 levels are associated with a more pronounced future eGFR decline. In particular, patients with uDKK3 levels ≥400 pg/ml in comparison to patients with baseline uDKK3 levels <400 pg/ml showed a statistically significant difference in eGFR drop.

  1. Line 220, please mention other analysis that you would recommend to perform not only limited to sample size 

Answer: We agree with the reviewer, that the biomarker DKK3 open a wide spectrum of possibilities for renal research activities. We therefore extended the scientific prospect in respect to uDKK3. In addition to the required further studies investigating the effects of antihypertensive therapies on renal outcome in dependence of uDKK3 levels, we extended this section within the revised version of the manuscript.

(…As uDKK3 might identify patients at high risk for CKD progression, it is of interest if intensification of nephroprotective therapy in dependence of uDKK3 levels might reduce CKD progression. Further studies are needed to investigate if uDKK3 may serve as a biomarker to improve the management of patients with various kidney diseases. Moreover, studies investigating the association between uDKK3 and the development of renal endpoints would be of special interest to determine if uDKK3 can predict relevant clinical course…)

Reviewer 2 Report

The authors have studied the utility of urinary DKK3 (uDKK3) to predict kidney function decline (i.e. loss of eGFR) in 31 patients with resistant hypertension randomized to two different treatment regimens. Patients with an average eGFR loss ≥3 ml/min/1.73 m2/year (n=13) had significantly higher uDKK3 levels than those with less kidney function decline. Moreover, patients with uDKK3 levels ≥400 pg/mg creatinine (n=11) showed a significant higher eGFR loss (-6.4±4.7 vs. 0.0±7.6 ml/min/1.73m2/year, p=0.0172). They concluded that in patients with resistant hypertension high levels of uDKK3 are associated with a larger short-term eGFR loss.

The following points should be addressed:

1) How many uDKK3 measurements were available in the individual patients? Was variability assessed; i.e. are data on the coefficient of variation (CV) for the DKK3 measurement available? Similar considerations apply to the eGFR measurement, are consecutive measurements available to plot individual eGFR trajectories in relation to baseline uDKK3?

2) CKD stage should be presented in Table 1, not in the text. The other data must not be repeated in the text and table.

3) Patients with high uDKK3 levels had lower baseline eGFR. Can the authors exclude an intrinsic propensity for CKD progression independent of DKK3 in these patients? Could the high uDKK3 in patients with low eGFR also be a result of accumulation of DKK3 in blood and thus consecutively higher urinary "load" (assuming that DKK3 is secreted not only in the urine but also in the blood by tubular cells)?

4) An additional possibility to present CKD progression in rather small studies without hard renal endpoints is to show how many patients reached a higher CKD stage in relation to baseline uDKK3. Is this information available?

5) There was a much larger decrease in systolic office and ambulatory blood pressure in patients with baseline uDKK3 >400 pg/mg creatinine compared to those with lower uDKK3. A larger reduction in blood pressure should act nephroprotective. So, the prediction power of uDKK3 might be actually "masked" by this imbalance in blood pressure control in the two groups. Could the authors comment on this?

Author Response

Please see also the attachment.

Reviewer 2

Comments and Suggestions for Authors

The authors have studied the utility of urinary DKK3 (uDKK3) to predict kidney function decline (i.e. loss of eGFR) in 31 patients with resistant hypertension randomized to two different treatment regimens. Patients with an average eGFR loss ≥3 ml/min/1.73 m2/year (n=13) had significantly higher uDKK3 levels than those with less kidney function decline. Moreover, patients with uDKK3 levels ≥400 pg/mg creatinine (n=11) showed a significant higher eGFR loss (-6.4±4.7 vs. 0.0±7.6 ml/min/1.73m2/year, p=0.0172). They concluded that in patients with resistant hypertension high levels of uDKK3 are associated with a larger short-term eGFR loss.

The following points should be addressed:

1) How many uDKK3 measurements were available in the individual patients? Was variability assessed; i.e. are data on the coefficient of variation (CV) for the DKK3 measurement available? Similar considerations apply to the eGFR measurement, are consecutive measurements available to plot individual eGFR trajectories in relation to baseline uDKK3?

Answer: We thank the reviewer for this deep critical analysis of the manuscript and want to answer these two questions raised by the reviewer.

  1. A) uDKK3 levels were determined exclusively once within the first available spot urine at baseline or up to a period of 6 months in duplicate, using a commercial ELISA kit (ReFiNE Dkk3 ELISA, DiaRen UG, Homburg (Saar), Germany). According to manufacturer information coefficient of variation (CV) for intra-assay variability of repeated urine sample measurements are 3.1 % in the lower detection range (at 488 pg DKK3/ml urine) and 3.5 % in the upper detection range (at 1472 pg DKK3/ml urine). The values for inter-assay test variability are 4.7% in the lower detection range and 5.1% in the higher detection range. For the present analysis all samples were measured on one ELISA plate. We included an according paragraph with CV within the revised version of the manuscript.

Methods

(…According to manufacturer information coefficient of variation for intra-assay variability of repeated urine sample measurements are 3.1 % in the lower detection range (at 488 pg DKK3/ml urine) and 3.5 % in the upper detection range (at 1472 pg DKK3/ml urine). The values for inter-assay test variability are 4.7% in the lower detection range and 5.1% in the higher detection range. For the present analysis all samples were measured on one ELISA plate….)

B)

For the present analysis, the latest eGFR follow-up was evaluated. As indicated within the methods section, with a median of 24 months (IQR 24-24) the vast majority of patients (87%) had latest follow-up at months 24 and only 4 patients (13%) had latest follow-up at month 12. For the present analysis, BP and eGFR were evaluated in follow-up at month 12 and month 24 and the latest follow-up value was evaluated.

Methods

(… For analysis, the latest follow-up values for BP and eGFR were used (evaluation time month 12 and 24…)

(…uDKK3 levels were determined exclusively once within the first available spot urine at baseline or up to a period of 6 months…)

Results

(…In particular, of the 31 analyzed patients, 4 patients (13%) had latest follow-up with determination of eGFR and BP at months 12 and 27 patients (87%) at months 24….)

Moreover, we could provide eGFR trajectories for each patients stratified to baseline DKK3-levels. If requested by the reviewer, we could add this additional figure with individual eGFR trajectories in relation to baseline uDKK3 to the supplement section.

  1. A) B)

Figure S1 individual eGFR trajectories in relation to baseline uDKK3

2) CKD stage should be presented in Table 1, not in the text. The other data must not be repeated in the text and table.

Answer: We agree with the reviewer, that existing duplications needs to be reduced. We therefore provide baseline characteristics exclusively within table 1 and and eGFR categories in the new table 4 which includes reviewer’s suggestion point 4.

3) Patients with high uDKK3 levels had lower baseline eGFR. Can the authors exclude an intrinsic propensity for CKD progression independent of DKK3 in these patients?

Could the high uDKK3 in patients with low eGFR also be a result of accumulation of DKK3 in blood and thus consecutively higher urinary "load" (assuming that DKK3 is secreted not only in the urine but also in the blood by tubular cells)?

Answer: We thank the reviewer for these two very interesting questions.

  1. Lower eGFR values in groups with higher uDKK3 levels as observed in the present study were already shown in patients with CKD. (e.g. Zewinger S et al. JASN 2018). The present data are therefore in line with previous reports.

As indicated by the reviewer, baseline eGFR is an established biomarker for prediction of CKD progression. However, in a large study with 1.7 million participants from 35 cohorts with 12,344 ESRD events, CKD progression was highly variable even in patients within the same eGFR category (Coresh et al JAMA 2014). Recently published results could demonstrate, that even after adjustment for potential confounders including baseline eGFR, uDKK3 remained an independent indicator of eGFR decline within the following 12-month period. (Zewinger et al. JASN 2018). Compared with eGFR or albuminuria alone, assessment of urinary DKK3 significantly improved prediction of short-term loss of kidney function. (Zewinger et al. JASN 2018)

An intrinsic propensity for CKD progression, due to reduced eGFR in patients with baseline uDKK3≥400 pg/mg creatinine, independent of uDKK3 in the present patients cannot be fully excluded. However, lack of correlation between baseline eGFR and change in eGFR makes it rather unlikely.

Results

(…There was no significant correlation of baseline eGFR with the change of eGFR at months 12 (r=-0.041, p=0.38) and months 24 (r=0.183, p=0.36)…)

Discussion

(… An intrinsic propensity for CKD progression, due to reduced baseline eGFR in patients with baseline uDKK3 ≥400 pg/mg creatinine, independent of uDKK3 in the present patients cannot be fully excluded. However, lack of correlation between baseline eGFR and change in eGFR makes it rather unlikely. Moreover, recently published results could demonstrate, that even after adjustment for baseline eGFR, uDKK3 remained an independent indicator of further eGFR decline [16]. Compared with eGFR or albuminuria alone, assessment of urinary DKK3 significantly improved prediction of further eGFR decline [16]. ….)

  1. The second part of the question raised by the reviewer was extensively discussed in a review by a leading DKK3 research group (Schunk SJ, Speer T, Petrakis I and Fliser D NDT 2021)

Since DKK3 is expressed also in other organs beside the kidneys, DKK3 can be detected within the plasma (Niehrs C et al. Oncogene 2006; Grone EF et al. Pflugers Arch Eur J Physiol 2017). Through substantial glycosylation DKK3 reaches a molecular weight of 60–70 kDa (Zhang K et al. International Journal of Oncology 2010), which makes a glomerular filtration of DKK3 in a great extend within an intact glomerulum unlikely.

Moreover, Schunk et al. stated, that in a yet unpublished study, they did not find a significant correlation between DKK3 in plasma and urine (Schunk SJ et al. NDT2021).

Urinary concentrations of DKK3 is only correlated with albuminuria in patients with CKD and not in subjects from the general population (Zewinger Set al. JASN 2018). The correlation coefficient for urinary DKK3 and albuminuria was 0.258 indicating only a weak correlation. In the CARE FOR HOME trial, a study comprising 575 CKD patients of various aetiologies, higher albuminuria (>300 mg/g creatinine) was associated with higher urinary DKK3 levels as compared with subjects with urinary albumin excretion <30 mg/g. (Zewinger Set al. JASN 2018)

However, in approximately 50% of the subjects with higher grade albuminuria, urinary DKK3 levels remained low (i.e. <1000 pg/mg creatinine) and in some patients even below the detection limit. (Zewinger Set al. JASN 2018) Therefore, it remains unclear why plasma DKK3 may not cross the glomerular barrier though the molecular mass of plasma DKK3 is similar to albumin. An explanation would be a potential interaction of plasma DKK3 with circulating plasma components or the formation of high molecular weight complexes (Schunk SJ et al. 2023).

Though of special interest, the second question raised by the reviewer cannot answered finally at the current time. Future studies are necessary to explore the detailed route of urinary and plasma DKK3 during kidney injury. We will discuss this interesting issue briefly within the revised version of manuscript.

Discussion

(…Through substantial glycosylation DKK3 reaches a molecular weight of 60–70 kDa (Zhang K et al. International Journal of Oncology 2010), which makes a glomerular filtration of DKK3 in a great extend within an intact glomerulum unlikely. Though of special interest, the question if high uDKK3 in patients with low eGFR may also be a result of accumulation of DKK3 in blood and thus consecutively leads to higher urinary "load" cannot answered finally at the current. Future studies are necessary to explore the detailed route of urinary and plasma DKK3 during kidney injury….)

4) An additional possibility to present CKD progression in rather small studies without hard renal endpoints is to show how many patients reached a higher CKD stage in relation to baseline uDKK3. Is this information available?

Answer: We thank the reviewer for this interesting hint for the analysis of CKD progression and we provide further analysis within the revised version of the manuscript. We included table 4 with the relevant information. Indeed, patients with uDKK3 levels ≥400 pg/mg creatinine at baseline experience more often a decrease in eGFR category compared to patients with uDKK3 <400 pg/mg creatinine (p=0.049).

Table 4. Baseline and changes of eGFR categories in respect to baseline uDKK3

Baseline

Latest Follow-up

eGFR categories (ml/min/1.73m2)

Patients with baseline uDKK ≥400 pg/mg creatinine (n=11)

Patients with baseline uDKK <400 pg/mg creatinine (n=20)

p-Value

Patients with baseline uDKK ≥400 pg/mg creatinine (n=11)

Patients with baseline uDKK <400 pg/mg creatinine (n=20)

p-Value*

>90

2 (18%)

7 (35%)

0.0304

1 (9%)

8 (40%)

0.0808

60-89

4 (36%)

8 (40%)

3 (27)

6 (30%)

30-59

1 (9%)

5 (13%)

2 (18%)

5 (25%)

15-29

4 (36%)

0 (0%)

3 (27%)

1 (5%)

<15

0 (0%)

0 (0%)

2 (18%)

0 (0%)

Decrease in eGFR category

-

-

6 (55%)

4 (20%)

0.049

P-values are based on the comparison of patients with an uDKK3 level more and less than 400 pg/mg creatinine (# baseline and * latest follow-up) values.

Results

(…Table 4 shows eGFR categories at baseline and at latest follow-up in respect to baseline uDKK3. Deterioration of eGFR categories from baseline to latest follow-up occurred in 6 patients (55%) with DKK3≥400 pg/mg creatinine and in 4 patients (20%) with DKK3 DKK3<400 pg/mg creatinine (p=0.049).…)

Methods:

 (…Categorical variables were compared using the Chi-Square test…)

5) There was a much larger decrease in systolic office and ambulatory blood pressure in patients with baseline uDKK3 >400 pg/mg creatinine compared to those with lower uDKK3. A larger reduction in blood pressure should act nephroprotective. So, the prediction power of uDKK3 might be actually "masked" by this imbalance in blood pressure control in the two groups. Could the authors comment on this?

Answer: We thank the reviewer for this deep critical analysis of our data. Though there was a numerical difference of the amount of BP reduction between patients with uDKK3> 400 pg/mg creatinine compared to those with lower uDKK3, this did not reach statistical significance. Therefore, discussion of this external effect of different BP response on prediction power of uDKK3 remains speculative. We included an according paragraph within the revised version of the manuscript.

Discussion

(…The numerically greater BP reduction in patients with higher uDKK3-levels after a median follow-up of 24 months (IQR24-24), might have contribute to a reduction in eGFR decline in this group. Considering the neprhoprotective effects of this greater BP reduction, the prediction power of uDKK3 might be actually even "masked" by this imbalance in BP control in the two groups….)

Round 2

Reviewer 1 Report

The manuscript has considerably improved and below please find minor comments

Line 28: the authors should specify the totals of each eGFR loss group and DKK3, after the cut-off points instead of lines 30 and 33.

Line 30: Since high and low groups are defined previously, you can omit the specific cut-off point and report directly the DKK3 values/medians per each group. 

Line 32: the Mann-whitney U and t-test specifications should be added before the results part of the abstract. I suggest adding a sentence about it between lines 28 and 29, after the specification of follow-up periods. Same for the Spearman correlation.  

Line 36: you can remove the "So" at the beginning of the sentence 

Line 212: Table 4 title can specify last follow-up change rather than baseline 

Author Response

Please see also the attachment

Reviewer 1

The manuscript has considerably improved and below please find minor comments

Line 28: the authors should specify the totals of each eGFR loss group and DKK3, after the cut-off points instead of lines 30 and 33.

Answer: According to the reviewer’s suggestion, we specified the according values directly after the cut-off point to give a better context for the reader

Line 30: Since high and low groups are defined previously, you can omit the specific cut-off point and report directly the DKK3 values/medians per each group. 

Answer: As suggested by the reviewer we omit the specific cut-off point and directly report the DKK3 values.

Line 32: the Mann-Whitney U and t-test specifications should be added before the results part of the abstract. I suggest adding a sentence about it between lines 28 and 29, after the specification of follow-up periods. Same for the Spearman correlation.  

Answer: As suggested by the reviewer we added statistical specification with the revised abstract.

Line 36: you can remove the "So" at the beginning of the sentence 

Answer: We removed the “So” within the revised version of the manuscript.

Line 212: Table 4 title can specify last follow-up change rather than baseline 

Answer: As indicated by the reviewer we changed the title of table 4 accordingly.
